# The Effect of COVID-19 on Arterial Stiffness and Inflammation: A Longitudinal Prospective Study

**DOI:** 10.3390/v17030394

**Published:** 2025-03-11

**Authors:** Jhony Baissary, Ziad Koberssy, Jared C. Durieux, Ornina Atieh, Joviane Daher, Kate Ailstock, Danielle Labbato, Theresa Foster, Michael A. Rodgers, Alexander Merheb, Nicholas T. Funderburg, Grace A. McComsey

**Affiliations:** 1School of Medicine, Case Western Reserve University, Cleveland, OH 44106, USA; jhony.baissary@case.edu (J.B.); ziad.koberssy@case.edu (Z.K.); ornina.atieh@case.edu (O.A.); joviane.daher@case.edu (J.D.); 2Clinical Research Center, University Hospitals Cleveland Medical Center, Cleveland, OH 44106, USA; jared.durieux@uhhospitals.org (J.C.D.); danielle.labbato@uhhospitals.org (D.L.); theresa.foster@uhhospitals.org (T.F.); michael.rodgers@uhhospitals.org (M.A.R.); 3Division of Medical Laboratory Science, School of Health and Rehabilitation Sciences, The Ohio State University, Columbus, OH 43210, USA; kate.ailstock@osumc.edu (K.A.); nicholas.funderburg@osumc.edu (N.T.F.); 4Harvard Faculty of Arts and Sciences, Harvard University, Cambridge, MA 02138, USA; amerheb@college.harvard.org

**Keywords:** COVID-19, inflammation, arterial stiffness, endothelial function, metabolic markers, gut integrity

## Abstract

Data are limited for assessing the effect of COVID infection on endothelial function, pre- and post-pandemic. The objective of this study was to assess changes in pre-pandemic cardiovascular parameters after COVID-19 infection. This prospective cohort study used EndoPAT2000 Itamar Medical Ltd., Caesarea, Israel, to measure the augmentation index (AI; arterial elasticity) and reactive hyperemic index (RHI; endothelial function). Markers of endothelial function, inflammation, and gut integrity were collected at pre- and post-pandemic visits. COVID-negative and COVID-positive participants were matched on pre-pandemic covariates, and AI ≥ 5.0 was defined as having worse AI. Among the 156 participants, 50% had documented COVID-19 infection. Groups were balanced (*p* > 0.05) on pre-pandemic characteristics. Increases in oxLDL (*p* = 0.03) were observed in the COVID-positive group, and COVID infection had a negative effect on inflammatory markers (sVCAM-1, sTNF-RI, sTNF-RII, sCD14) and gut integrity (I-FABP, BDG) compared to COVID-negative participants (*p* < 0.05). There was a 16.7% (*p* = 0.02) increase in the proportion of COVID-positive participants with AI ≥ 5.0, without a significant change (*p* = 0.09) among the COVID-negative group. COVID-positive status, female sex, and higher IL-6 and sCD163 were associated (*p* < 0.05) with an increase in having worse AI. COVID infection is independently associated with arterial stiffness. For COVID survivors, female sex and higher markers of inflammation were associated with arterial stiffness.

## 1. Introduction

Since the emergence of the SARS-CoV-2 virus in December 2019, the global impact of the coronavirus disease COVID-19 has affected millions of individuals worldwide, and the World Health Organization (WHO) declared it a pandemic in March 2020 [1]. The manifestations of COVID-19 extend beyond the acute respiratory illness to include long-term effects involving multiple systems within the human body [2], a condition referred to as long COVID [3]. The mechanisms underlying long COVID are not fully understood. Still, studies suggest that chronic inflammation plays a significant role in the pathogenesis of these persistent symptoms and health risks [4,5].

In this context, the SARS-CoV-2 invades the host cell by binding its spike protein to the angiotensin converting enzyme 2 (ACE2) in the cell surface; this receptor is overexpressed in the endothelium, lungs, and intestines [6,7]. Zhang et al.’s study has shown that the SARS-CoV-2 infection induces long-lasting transcriptional changes in immune cells, consistent with an amplified pro-inflammatory state [8]. Direct infection of endothelial cells by SARS-CoV-2 and the subsequent cytokine storm produced as a result of infection may influence endothelial cell function [9].

The acute phase of SARS-CoV-2 infection has many effects on the cardiovascular system, including thrombosis formation, myocarditis, arrhythmias, and endothelial damage [10,11]. Even mild cases of COVID-19 disease are associated with subclinical endothelial and cardiovascular injury in the acute phase [12]; however, examining sustained vascular injury post-COVID-19 remains understudied.

Oxidized low-density lipoprotein (oxLDL) is a biomarker increased in the post-acute phase of COVID-19 that correlates with persistent symptoms of long COVID [13]. oxLDL plays a critical role in endothelial damage, the induction of inflammation, thrombus formation, and plaque destabilization. This is attributed to oxLDL’s involvement in the phenotypic transformation of macrophages into foam cells and the chemotactic and immunogenic properties of this molecule [14,15,16].

The association between arterial stiffness and the long-term effect of SARS-CoV-2 has been explored, but many of these studies were small, cross-sectional studies investigating arterial stiffness after COVID-19 disease without pre-COVID baseline measures. Ratchford et al. conducted a cross-sectional study of 37 participants, showing that arterial function is impaired 3 to 4 weeks after COVID-19 infection [17]. Another prospective cohort study showed that even mild cases of SARS-CoV-2 infection could be associated with widespread pathological processes in the vessels three months after COVID-19 [12]. We have previously shown that long COVID symptoms were associated with arterial stiffness [18] and that females are the most affected by arterial stiffness after COVID-19 infection [19]. Unlike previous studies, our work was a cohort in nature with pre-infection measures.

To our knowledge, this is the first cohort to assess prospectively the effect of SARS-CoV-2 infection on arterial stiffness using peripheral arterial tonometry after a median of 15 months post-infection compared to these measurements in a group of controls who never had COVID-19 infection. Also, this study evaluates the relationships among endothelial function changes and levels of oxLDL, inflammatory, and gut integrity markers pre- and post-pandemic.

## 2. Materials and Methods

### 2.1. Study Design and Population

This prospective cohort study was conducted at the University Hospitals Cleveland Medical Center (UHCMC), Cleveland, Ohio, from July 2021 to May 2024. We enrolled participants aged 18 years and older with a documented pre-pandemic endothelial function assessment using the EndoPAT^®^-2000 device (Itamar Medical Ltd., Caesarea, Israel), which had been obtained as part of prior studies conducted at the Metabolic Research Center at UHCMC. This current study’s primary exposure of interest was COVID-19 infection (COVID positive), confirmed through either a nucleic acid amplification test (NAAT), positive serology (nucleocapsid antibody, or spike antibody if unimmunized), or any Food and Drug Administration (FDA)-approved SARS-CoV-2 antigen test. The non-exposed control group (COVID negative) consisted of individuals with available EndoPAT testing obtained in the same timeframe pre- and post-pandemic using the same technique and machine, but without documented COVID-19 infection. This control group was propensity matched to the exposed group based on age and sex at birth. The EndoPAT-2000 we used was an FDA-approved instrument to evaluate endothelial function by measuring the reactive hyperemic index (RHI) and the augmentation index (AI), reflecting endothelial function and arterial elasticity, respectively. Based on the availability of pre-pandemic EndoPAT, a total of 156 participants met the eligibility criteria for inclusion and underwent a post-pandemic EndoPAT testing by adequately trained technicians. In addition to EndoPAT, blood samples were collected for the measurement of specific metabolic, gut integrity, and inflammatory markers and oxLDL. All enrolled participants completed a baseline clinical assessment, detailed medical and COVID-19 history taking, and specific metabolic, gut integrity, and inflammatory marker measurements.

### 2.2. Study Measurements

#### 2.2.1. Baseline Characteristics of Participants

Well-trained healthcare professionals applied standardized questionnaires to collect comprehensive data on participants at their first EndoPAT visit, including demographic characteristics, lifestyle factors including tobacco use, and medical history.

#### 2.2.2. Biomarkers

Participants were instructed to fast for at least 12 h before blood collection. All blood samples were sent to a CLIA-certified laboratory for metabolic measurements, including triglycerides (TG), total cholesterol, non-high-density lipoprotein (non-HDL), and glycated hemoglobin or hemoglobin A1C (HbA1c).

Additional plasma samples were promptly stored at −80 °C within 2 h and were shipped to the Funderburg laboratory, Ohio State University, for the measurement of biomarkers by enzyme-linked immunosorbent assay (ELISA). Samples were thawed once and run in batch. Levels of high-sensitivity C-reactive protein (hsCRP), the monocyte/macrophage activation markers soluble CD14 and CD163 markers (sCD14 and sCD163), interleukin (IL-6), tumor necrosis factor receptor 1 and 2 (sTNF-RI and sTNF-RII), intercellular adhesion molecule (sICAM-1), and vascular cell adhesion molecule (sVCAM-1) were measured using kits from R&D Systems (Minneapolis, MN, USA); D-dimer and oxidized low density lipoprotein were measured using kits from Diagnostica Stago (Parsippany, NJ, USA) and Mercodia (Uppsala, Sweden), respectively. The gut integrity marker intestinal fatty acid binding protein (I-FABP, R&D Systems), gut permeability marker zonulin (Immunodiagnostik AG, Heidelberg, Germany), and the microbial translocation markers lipopolysaccharide-binding protein (LBP, R&D Systems) and β-D-Glucan (BDG, MBS756415, MyBioSource, San Diego, CA, USA) were also measured.

#### 2.2.3. Vascular Function

Participants were required to avoid exercise for at least 4 h and refrain from consuming caffeinated beverages, tobacco, vitamins, or medications that could potentially interfere with vascular function for at least 8 h prior to undergoing EndoPAT testing.

In this study, we utilized the EndoPAT^®^-2000 device (Itamar Medical) as a rapid, non-invasive tool to evaluate the endothelial vasodilator function. This is typically evaluated by examining the ability of endothelial cells to produce vasodilation signals (prostacyclin, nitric oxide …) in response to hyperemia [20]. An indirect assessment of the generation of these signals could be obtained by studying changes in peripheral arterial tone (PAT) during hyperemia-induced vasodilation.

The EndoPAT device consists of two plethysmographic probes placed on the same finger of each hand to record volume changes with each arterial pulse, creating a beat-to-beat arterial pulse wave amplitude (PWA). These pneumatic fingertip probes contain inflatable air cushions that enhance measurement sensitivity by applying a uniform, near-diastolic pressure field across the finger. This will prevent venous pooling that might cause veno-arteriolar reflex vasoconstriction and partially unload arterial wall tension. The pressure change signals are transmitted from the fingertip probes to a computer, where the signals are filtered, amplified, displayed, and saved for future use [21].

The PAT study was conducted in a comfortable and thermoneutral environment, with the patient seated and both hands positioned at the same level. A blood pressure cuff is applied to the non-dominant upper arm (study arm), while the contralateral arm serves as a control. Recordings are taken simultaneously from both arms throughout this study. The PAT study consists of three phases: baseline, occlusive, and hyperemia. During the baseline phase, the probes record baseline PWA values. Subsequently, the blood pressure cuff on the study arm is inflated to 60 mmHg above systolic pressure (not less than 200 mmHg) for 5 min. During occlusion, signals are absent from the study arm but continue from the control arm. The cuff is then deflated to induce reactive hyperemia and assess PWA during this phase. This process generates a reactive hyperemic index (RHI) from changes in baseline PWA to post-occlusion PWA in the occluded arm. The RHI of the study arm is corrected for corresponding changes in PWA relative to baseline in the contralateral, non-occluded arm to minimize the influence of non-endothelial dependent systemic changes. A normal index is greater than 1.67, while an abnormal index is 1.67 or less [22].

The augmentation index (AI) is calculated from PAT pulses recorded during the baseline period and corrected to a standard heart rate of 75 beats per minute (AI 75). Lower augmentation index values indicate better arterial elasticity.

### 2.3. Statistical Analysis:

Characteristics of study participants were described using mean +/− standard deviation (SD) or median and interquartile range (IQR) for continuous variables and frequency (n) and percentage (%) for categorical variables; the Shapiro–Wilk test was used to test the normality assumption. Differences between groups were computed using an independent *t*-test or non-parametric, Mann–Whitney continuous variables and chi-square or Fisher’s exact for categorical variables. COVID-negative participants were 1:1 propensity score matched to COVID-positive participants on pre-pandemic covariates that included age, sex, and the number of days between visits using greedy nearest neighbor matching. AI was dichotomized using the overall sample median [5.0 (IQR: −4, 18.5)] and worse arterial elasticity was defined by AI ≥ 5.0. We used longitudinal linear mixed models with random intercept to test whether within and between-subject changes in markers of endothelial function, inflammation, gut permeability, and monocyte/macrophage activation were associated with COVID infection. We used generalized linear mixed models with a binary distribution and adaptive Gaussian quadrature to estimate the effect of COVID infection on the likelihood of having worse AI (≥5.0). Adjusted models included pre-pandemic age, sex, race, smoking status, non-HDL, and BMI. Markers of inflammation were modeled independently of each other and log transformations were used to reduce the error variance. All analyses were conducted using SAS 9.4 (SAS Inc., Cary, NC, USA) and *p*-values less than alpha <0.05 were considered statistically significant.

## 3. Results

### 3.1. Characteristics of Participants

Overall (n = 156), 50% of participants were COVID positive and 50% were COVID negative (Table 1). Among the COVID-positive group pre-pandemic, the median age was 46.9 (IQR: 30.2, 55.1) years, 40.3% were female sex, the average BMI was 28.8 ± 6.3 kg/m^2^, and the median number of days between pre- and post-pandemic visits was 1105 (IQR: 640, 1617). Study groups were balanced (*p* > 0.05) on all pre-pandemic characteristics except smoking status (*p* = 0.01), sTNF-RI (*p* = 0.03), sTNF-RII (*p* < 0.001), and sI-CAM-1 (*p* < 0.001).

### 3.2. Changes in Metabolic, Inflammation, Monocyte/Macrophage Activation, and Gut Permeability Markers from Pre-Pandemic to Post-Pandemic Timepoints

Over the observation period (Table 2), a reduction in total cholesterol [−11.9 mg/dL (*p* = 0.01)], non-HDL cholesterol [−10.8 mg/dL (*p* = 0.03)], and BDG (−89.9 pg/mL; *p* = 0.01) was observed while levels of inflammatory marker oxLDL (+11.6; *p* = 0.03) and monocyte/macrophage activation marker soluble CD14 (+186.1; *p* = 0.01) increased among the COVID-positive group. No significant (*p* > 0.05) changes in cholesterol, non-HDL, BDG, oxLDL, or sCD14 were observed among the COVID-negative participants. Comparing the changes in markers between groups, COVID infection increased the inflammation (sVCAM-1, sTNF-RI, sTNF-RII), monocyte/macrophage activation (sCD14), and gut integrity (I-FABP, BDG) compared to COVID-negative participants (*p* < 0.05). However, there was insufficient evidence (*p* > 0.05) to suggest that changes in BMI, IL-6, hsCRP, and D-dimer were a result of COVID infection

### 3.3. Endothelial Function and Arterial Elasticity

Overall, there was a 16.7% (*p* = 0.02) increase in the proportion of COVID-positive participants with AI ≥ 5.0 (*p* = 0.02) without a significant change (−2.6%; *p* = 0.09) in the proportion of COVID-negative participants with AI ≥ 5.0. However, neither RHI (−0.1 ± 0.7; *p* = 0.3) nor the proportion of COVID-positive participants with RHI ≤ 1.67 (4.1%; *p* = 0.6) did not significantly change, while there was a 7.3% increase (*p* = 0.04) in the proportion of COVID-negative participants with RHI ≤ 1.67.

Outlined in Table 3, the COVID-positive group was more than four times more likely to have worse arterial elasticity (AI ≥ 5.0) compared to the COVID-negative group [unadjusted odds ratio (uOR): 4.5 (95% CI: 1.6, 13); *p* = 0.02]. Female sex was four times more likely to have worse AI compared to the male sex [uOR: 4.0 (95% CI: 2.4, 6.9); *p* < 0.0001] and every one-year increase in pre-pandemic age increased the odds of having worse AI by 1.1 (95% CI: 1.1, 1.2; *p* < 0.0001). Similarly, more than a two-fold increase in the likelihood of having worse AI was observed for every unit increase in inflammatory markers D-dimer [uOR: 2.1 (95% CI: 1.4, 3.2); *p* = 0.0002], sTNF-RI [uOR: 3.3 (95% CI: 1.2, 9.1); *p* = 0.03], and sTNF-RII [uOR: 4.5 (95% CI: 1.6, 12.7); *p* = 0.004. After adjusting (aOR) for pre-pandemic age, sex, race, smoking status, non-HDL, and BMI, the odds of having worse AI remained 4 times higher in the COVID-positive group compared to the COVID-negative group and every unit increase in soluble CD163 [aOR: 2.8 (95% CI: 1.5, 5.2); *p* = 0.001] was associated with nearly a three-fold increase in the odds of having worse AI. IL-6 [aOR: 1.4 (95% CIs: 1.04, 1.9); *p* = 0.03] was associated with an increased odds of having worse AI. There was insufficient evidence (*p* > 0.05) to suggest that being a current smoker, BMI, non-HDL, and race increased the risk of having worse AI.

## 4. Discussion

In this prospective cohort study, we investigated the effect of COVID-19 infection on endothelial function and arterial stiffness by comparing the RHI and AI post-pandemic to pre-pandemic measurements performed on the same group using the same standardized technique and machine. We also compared AI and RHI changes to a demographically matched group who never had COVID-19 but had a pre-pandemic and a follow-up (post-pandemic) EndoPAT. While RHI changes were not different between groups by COVID status, nor different post-COVID compared to pre-COVID, the AI results showed an effect of COVID status, with overall COVID survivors showing worsening arterial stiffness over time and compared to the control group who never had COVID.

There was evidence (*p* < 0.05) of an increase in sCD14 within the COVID-positive group and a decrease in the COVID-negative group between pre- and post-pandemic visits; our data suggest that this change (as with VCAM-1, TNFr-I, and TNFr-II) is attributable to COVID infection. Monocyte activation (sCD14) is a likely source of the heightened inflammation in chronic illnesses. Our results also showed a non-significant increase in hsCRP in the COVID-negative group, but one possible explanation is that the COVID-negative group had nearly two times the proportion of current smokers.

Additionally, this study’s results emphasize the effect of sex and age on arterial stiffness, where female and older participants showed worse AI compared to other COVID-positive subgroups. Furthermore, the results highlight the correlation between inflammation and worsening arterial stiffness among COVID survivors. Our study showed no significant correlation between the arterial function changes and other long COVID symptoms.

Arterial stiffness is implicated in many cardiovascular and systemic diseases. Previous studies have implicated COVID-19 in many cardiovascular diseases, significantly increasing the risk of major adverse cardiovascular events, including myocardial infarction, stroke, and all-cause mortality [23]. Moreover, previous research found that COVID may trigger new-onset hypertension [24], and arterial stiffness is probably the underlying pathophysiology of high blood pressure among COVID survivors [25].

The results of Ratchford et al., a cross-sectional study showing the difference in arterial function between a COVID-19 survivors’ group and a group who never had COVID-19, align with the results of our study, reporting higher arterial stiffness among the COVID+ group compared to the group who never had COVID. The time between COVID-19 infection and the measurement in the study was around 25 days, unlike our study, where it was around 15 months, supporting the long-term effect of SARS-CoV-2 on vascular function. In contrast to our study, carotid-femoral wave velocity cfPWV was used to assess the arterial stiffness [17]. In another study that used tonometry to assess arterial stiffness, Podrug et al. reported no increase in arterial stiffness 2–3 months post-COVID-19 infection. However, a 2–3 month follow-up is likely not long enough to see such an effect, and in our study with a 15-month follow-up after COVID-19, we found a significant increase in the proportion of participants with higher AI in the COVID+ group compared to pre-COVID assessment and significantly higher AI compared to the COVID-negative group [12].

A previous cross-sectional study completed in the metabolic research center at University Hospitals Cleveland Medical Center, Cleveland, Ohio, using peripheral arterial tonometry demonstrated a significantly higher AI among COVID-19 infected participants compared to the COVID-negative group with no difference in RHI, and AI was significantly higher among women and older participants. In the present study, we extend our observation in a prospective longitudinal setting where the same participants had baseline pre-pandemic measures [18].

In this study, arterial stiffness significantly improved among the COVID-negative group over time; this could be associated with increased health awareness during the pandemic and better cardiovascular risk factor control in terms of healthier lifestyles in some cases. Multiple studies have shown that a healthier lifestyle, better patient–physician communication, more sleep hours, increased vegetable consumption, and increased use of home exercise equipment were observed during the pandemic [26,27].

Our study found that inflammation, especially sVCAM-1, sTNF-RI, sTNF-RII, and oxLDL, increased significantly in the COVID survivors group compared to the group who never had COVID; this agrees with the results of other studies that show significantly higher inflammation and oxLDL in the COVID+ groups compared to COVID-negative groups [13,28]. Additionally, the results of a study completed in Germany reported chronic elevation in certain inflammatory marker among people with post-acute sequalae of SARS-CoV-2, which is consistent with the results of our study [29].

In the context of inflammation, osteopontin stands out as a potential key biomarker that warrants further investigation, especially given its association with patients experiencing post-acute sequelae of COVID-19 [30], since long COVID is linked to a heightened coagulable state in the acute hospital stay [31]. Therefore, it seems that osteopontin and hypercoagulability play a significant role in the long-term sequelae observed after COVID-19, one of which could be the worsening of arterial function post-infection seen in our study.

The results of our study shed light on the importance of following up with patients post-COVID infection to better control their cardiovascular risks and adopt new approaches to reduce complications secondary to worsening vascular health. Compliance with blood pressure control, smoking cessation, and a healthier diet should be monitored, especially in COVID-19 survivors who tend to have worse endothelial health.

Our study has several strengths that differentiate it from other studies, importantly having available pre-pandemic EndoPAT measures. Also, we have a comprehensive measure of inflammatory and gut markers that have not been performed in prior studies. However, we acknowledge that our study has some limitations, including the fact that the EndoPAT method indirectly measures arterial stiffness, and that the lack of peripheral arterial tonometry measurements during the acute phase of COVID-19 infection can make impossible the attribution of arterial stiffness worsening due to COVID-19 infection alone. Although we lacked sufficient data on the vaccination status of the participants, this variable is crucial and should be considered in future research.

## 5. Conclusions

In conclusion, this study shows that arterial stiffness worsened in the group of COVID survivors compared to pre-pandemic measures and compared to a group of participants who never had COVID. Women and older individuals seemed most affected. In addition, several inflammatory markers, but not gut markers, were associated with worsening arterial stiffness. Additional studies are needed to investigate the long-term cardiovascular effect of arterial stiffness caused by SARS-CoV-2 infection.

## Figures and Tables

**Table 1 viruses-17-00394-t001:** Characteristics of study participants pre-pandemic by COVID status.

		COVID Positive (n = 78)	COVID Negative (n = 78)
		mean ± std/median (IQR) or n (%)	
	Age (years) *	46.9 (30.2, 55.1)	52.4 (34.6, 59.3)
	Female Sex *	31 (40.3)	21 (26.9)
	Non-white Race **	41 (53.3)	45 (57.7)
	Current Smoker (Yes)	22 (28.9)	40 (51.9)
	Total weekly physical activity (mins)	2905 (1770, 4200)	2100 (960, 3180)
	Number of days between visits *	1105 (640, 1617)	829 (526, 1454)
	Number of days since COVID infection ***	451 (194, 939)	--
Metabolic Markers			
	BMI (kg/m^2^)	28.8 ± 6.3	27.4 ± 5.4
	HgbA1c (%)	5.5 ± 0.5	6.1 ± 2.2
	Cholesterol (mg/dL)	181.6 ± 37.7	170.7 ± 37.0
	non-HDL (mg/dL)	129.7 ± 34.3	124.8 ± 37.5
	Triglycerides (mg/dL)	120.1 ± 64.9	139.9 ± 93.7
Endothelial Function			
	Reactive Hyperemic Index	1.9 ± 0.6	1.6 (1.5, 2)
	Augmentation Index	5.7 ± 16.9	10.1 ± 16.6
Inflammatory Markers			
	IL-6 (pg/mL)	2.1 (1.4, 3.3)	2.9 (1.6, 3.9)
	sVCAM-1 (ng/mL)	718.6 (591.1, 864.9)	759.7 (616.3, 952.7)
	sTNF-RI (pg/mL)	926.3 (803.7, 1153.7)	1066.4 (801.6, 1278.6)
	sTNF-RII (pg/mL)	2324.7 (1889.5, 2655.6)	2653.5 (2234.4, 3778.4)
	hsCRP (ng/mL)	2368.4 (895.8, 5394)	2746.1 (1221.9, 6475.6)
	sI-CAM-1 (ng/mL)	221.9 (174.8, 285.6)	286.1 (227.4, 368.1)
	D-dimer (ng/mL)	366 (212.2, 503.6)	432.5 (273.9, 653.5)
	oxLDL (U/L) ****	51.6 (38.3, 73.6)	49.6 (38.5, 71.3)
Monocyte/Macrophage Activation Markers			
	sCD14 (ng/mL)	1518.6 (1304.2, 1938.9)	1683.5 (1396.1, 1943.2)
	sCD163 (ng/mL)	640.9 (453.7, 854.3)	689.1 (457.5, 1000.6)
Gut Markers			
	Zonulin (ng/mL)	1.7 (0.9, 4)	1.3 (0.8, 3.9)
	I-FABP (pg/mL)	1763.2 (1248.9, 2445.7)	2251.1 (1569.4, 3074.3)
	LBP (ng/mL)	17,456 (12,423, 23,396.6)	16,609.1 (12,066.5, 23,138.5)
	BDG (pg/mL)	192.9 (119.1, 367.1)	146.2 (93.1, 327.7)

Note: Groups were balanced (*p* > 0.05) on pre-pandemic characteristics except smoking status (*p* = 0.01), TNF-rI (*p* = 0.03), sTNF-rII (*p* < 0.001), and I-CAM-1 (*p* < 0.001). * Propensity score matching variables. ** Includes African American, Asian, Hispanic, and Other. *** Number of days from COVID infection to post-pandemic visit. **** Oxidized low-density lipoprotein (per 1000). Abbreviations: BMI = Body Mass Index; HbA1C = hemoglobin A1c; non-HDL = non-high-density lipoprotein; IL-6 = interleukin-6; sVCAM-1 = vascular cell adhesion molecule-1; sTNF-RI = tumor necrosis factor receptor-1; sTNF-RII = tumor necrosis factor receptor-2; hsCRP = high-sensitivity C-reactive protein; sI-CAM = intercellular adhesion molecule-1; oxLDL = oxidized LDL; sCD14 = soluble CD14; sCD163 = soluble CD163; I-FABP = intestinal fatty acid binding protein; LBP = lipopolysaccharide binding protein; BDG = β-D-glucan.

**Table 2 viruses-17-00394-t002:** Changes in biomarkers post-pandemic by COVID status.

		∆ (*p-Value*) *				*p-Value* **
		COVID Positive		COVID Negative		
		mean ± std or % ***				
Metabolic Markers						
	BMI (kg/m^2^)	4 ± 33.3	(0.3)	−0.2 ± 3.1	(0.7)	0.3
	HgbA1c (mmol/mol)	−0.1 ± 0.3	(0.2)	−0.1 ± 0.3	(0.4)	0.9
	Cholesterol (mg/dL)	−11.9 ± 30.3	(0.01)	−3.5 ± 42.3	(0.4)	0.2
	non-HDL (mg/dL)	−10.8 ± 27.2	(0.03)	−5.1 ± 43.6	(0.3)	0.5
	Triglycerides (mg/dL)	−5.3 ± 55.1	(0.5)	−0.9 ± 89.1	(0.9)	0.8
Endothelial Function						
	Reactive Hyperemic Index	−0.1 ± 0.7	(0.3)	−0.1 ± 0.6	(0.07)	0.6
	RHI ≤ 1.67	4.1	(0.6)	7.3	(0.04)	0.6
	Augmentation Index	2.7 ± 15.2	(0.1)	−3.2 ± 14.6	(0.06)	0.01
	AI ≥ 5	16.7	(0.02)	−2.6	(0.09)	0.1
Inflammatory Markers						
	IL-6 (pg/mL)	−1.3 ± 15.2	(0.6)	−0.5 ± 5.7	(0.4)	0.7
	sVCAM-1 (ng/mL)	74.4 ± 279.4	(0.2)	−96.7 ± 507.1	(0.8)	0.03
	sTNF-RI (pg/mL)	31 ± 285.9	(0.5)	−206.6 ± 358.0	<0.0001	<0.0001
	sTNF-RII (pg/mL)	121.3 ± 1001.1	(0.4)	−615.7 ± 1364.6	(0.01)	0.001
	hsCRP (ng/mL)	−550.2 ± 5774.7	(0.5)	162.5 ± 11,564.0	(0.9)	0.8
	sI-CAM-1 (ng/mL)	−18.7 ± 48.8	(0.2)	−35.1 ± 122.8	0.03	0.4
	D-dimer (ng/mL)	−602.7 ± 3377.5	(0.2)	−822.3± 4198.9	(0.1)	0.7
	oxLDL (U/L)	11.6 ± 34.6	(0.03)	−0.8 ± 38	(0.8)	0.1
Monocyte/Macrophage Activation Markers						
	sCD14 (ng/mL)	186.1 ± 452.1	(0.01)	−24.4± 688	(0.4)	0.04
	sCD163 (ng/mL)	684.6 ± 316.2	(<0.0001)	777.8 ± 433.9	(<0.0001)	0.1
Gut Permeability						
	Zonulin (ng/mL)	−1.4 ± 5.4	(0.1)	−0.9 ± 3.3	(0.1)	0.6
	I-FABP (pg/mL)	−417.5 ± 1509.6	(0.07)	643.9 ± 2273.5	(0.03)	0.01
	LBP (ng/mL)	−2973.8 ± 9562.3	(0.04)	−2683.7 ± 15,299.0	(0.2)	0.9
	BDG (pg/mL)	−89.9 ± 198.8	(0.01)	9 ± 247.7	(0.8)	0.04

* Within group change between pre-pandemic and post-pandemic. ** Change from pre-pandemic to post-pandemic between groups. *** % = proportion.

**Table 3 viruses-17-00394-t003:** Likelihood of having worse arterial elasticity *.

	Unadjusted		Adjusted **	
	OR (95% CIs)	*p*-Value	OR (95% CIs)	*p*-Value
COVID Status (+ vs. −)	4.5 (1.6, 13)	0.02	4.1 (1.4, 11.9)	0.02
Pre-pandemic Age (years)	1.1 (1.1, 1.2)	<0.0001	1.1 (1.1, 1.2)	<0.0001
Female Sex (vs. Male)	4 (2.4, 6.9)	<0.0001	4.4 (2.3, 8.9)	0.0003
Race (non-white vs. white)	1 (0.7, 1.6)	0.9	1.7 (0.9, 3.1)	0.5
Current Smoker (Y vs. N)	1.6 (0.5, 4.9)	0.4	1.7 (0.9, 3.1)	0.5
BMI (kg/m^2^)	1.5 (0.3, 7.4)	0.6	0.7 (0.2, 2.4)	0.6
non-HDL (mg/dL)	0.9 (0.3, 2.9)	0.8	0.7 (0.2, 2)	0.5
IL-6 (pg/mL)	1.8 (1.2, 2.6)	0.01	1.4 (1.04, 1.9)	0.03
D-dimer (ng/mL)	2.1 (1.4, 3.2)	0.0002	1.2 (0.8, 1.6)	0.3
sTNF-RI (pg/mL)	3.3 (1.2, 9.1)	0.03	2.5 (0.9, 6.7)	0.07
sTNF-RII (pg/mL)	4.5 (1.6, 12.7)	0.004	2 (0.8, 5)	0.2
sCD163 (ng/mL)	3.7 (1.7, 7.8)	0.001	2.8 (1.5, 5.2)	0.001

* Worse arterial elasticity = AI ≥ 5.0. ** Adjusted models included pre-pandemic age, sex, race, smoking status, non-HDL, and BMI. Markers of inflammation were modeled independently of each other.

## Data Availability

The data supporting this study’s findings are available on request from the corresponding author, GAM. The datasets presented in this article are not readily available because they are part of an ongoing study.

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
