# Peer review of "The Effect of COVID-19 on Arterial Stiffness and Inflammation: A Longitudinal Prospective Study"

_viruses, 2025, doi:10.3390/v17030394_

Round 1
Reviewer 1 Report
Comments and Suggestions for AuthorsÂ
In this manuscript the AA assess changes in pre-pandemic cardiovascular parameters after COVID infection.
The parameters assessed include inflammatory markers, endothelial and gut integrity, arterial elasticity, metabolic markers, monocytic activities.
The changes in metabolic, inflammation, monocyte activation and gut permeability markers from pre-pandemic and post-pandemic participants were evaluated.
According to the results obtained, the AA concluded that the arterial stiffness worsened in the group of COVID survivors.
Overall, this is a very useful study, which could help to prevent and manage the increased risk of cardiovascular events in frailty subjects with clinical infection with SARS-COV2.
Could be also important to perform a similar evaluation concerning the probability of an increased risk of stroke events.
Â
Author Response
Thank you for your insightful comments on my manuscript; they are greatly appreciated.
Reviewer 2 Report
Comments and Suggestions for Authors
I reviewed the viruses-3427095 mnuscript. The authors conducted an interesting study depicting that COVID-19 survivors exhibited a worsening arterial stiffness score compared to pre-pandemic measurements. Furthermore, they observed that COVID-19 status was independently associated with an increased propability of arterial stiffness. The strengths of the study are its prospective nature, the long follow-up period, the control group of non-COVID patients and the use of mixed effects model. However, certain limitations make this manuscript not suitable for scientific publication (at least in its current form). The most important ones are the following:
1) It is stated in the "discussion" section that "Our study found that inflammation, especially VCAM, TNF-RI, TNF-RII, and oxLDL increased significantly in the COVID survivors group compared to the group who never had COVID". However from the findings depicted in table 2 and the subsequent analysis the reader understands the exact oposite, which is also stated in the 'results' section; "Comparing the changes in markers between groups, COVID infection had a negative effect on inflammation VCAM, TNF-RI, TNF-RII..". The authors should clarify their findings and discuss them in relation to the findings of the study by Schultheiss et al. (Schultheiß C, Willscher E, Paschold L, Gottschick C, Klee B, Henkes SS, Bosurgi L, Dutzmann J, Sedding D, Frese T, Girndt M, Höll JI, Gekle M, Mikolajczyk R, Binder M. The IL-1β, IL-6, and TNF cytokine triad is associated with post-acute sequelae of COVID-19. Cell Rep Med. 2022 Jun 21;3(6):100663. doi: 10.1016/j.xcrm.2022.100663.).
2) The authors should provide thourough details about the mixed effect models (at least in a supplementary file).
3) The authors should clarify whether paired statistical tests were performed in order to identify whether changes in time (pre- and post-pamdemic differences) regarding the separate groups (COVID and non-COVID) were statistically significant. If this is not the case, the analysis should be repeated using paired tests.
4) In the "limitations" paragraph the authors should state that the EndoPAT method evaluates arterial stiffness indirectly.
5) The authors should explain in the 'limitations' paragraph how does the increased percentage of current smokers in the non-COVID group (compared to the COVID group) affect their analysis.
6) Since high plasma osteopontin has been associated with atherosclerosis (Kadoglou NPE, Khattab E, Velidakis N, Gkougkoudi E. The Role of Osteopontin in Atherosclerosis and Its Clinical Manifestations (Atherosclerotic Cardiovascular Diseases)-A Narrative Review. Biomedicines. 2023 Nov 29;11(12):3178. doi: 10.3390/biomedicines11123178.) and hypercoagulability tends to increase atherosclerosis (Loeffen R, Spronk HM, ten Cate H. The impact of blood coagulability on atherosclerosis and cardiovascular disease. J Thromb Haemost. 2012 Jul;10(7):1207-16. doi: 10.1111/j.1538-7836.2012.04782.x.), the authors should discuss their findings in relation to previous reports indicating that patients with serious post-acute COVID-19 sequelae were characterized by increased plasma osteopontin levels (Pappas AG, Eleftheriou K, Vlahakos V, Magkouta SF, Riba T, Dede K, Siampani R, Kompogiorgas S, Polydora E, Papalampidou A, Loutsidi NE, Mantas N, Tavernaraki E, Exarchos D, Kalomenidis I. High Plasma Osteopontin Levels Are Associated with Serious Post-Acute-COVID-19-Related Dyspnea. J Clin Med. 2024 Jan 10;13(2):392. doi: 10.3390/jcm13020392.) and that the patients with long COVID were characterized by hypercoagulable ROTEM (Loutsidi NE, Politou M, Vlahakos V, Korakakis D, Kassi T, Nika A, Pouliakis A, Eleftheriou K, Balis E, Pappas AG, Kalomenidis I. Hypercoagulable Rotational Thromboelastometry During Hospital Stay Is Associated with Post-Discharge DLco Impairment in Patients with COVID-19-Related Pneumonia. Viruses. 2024 Dec 14;16(12):1916. doi: 10.3390/v16121916.). Â
Apart from a few grammar/vocabulary errors the manuscript is generally well written
Author Response
Thank you for your insightful comments on my manuscript; they are greatly appreciated and have helped improve the work.
1) It is stated in the "discussion" section that "Our study found that inflammation, especially VCAM, TNF-RI, TNF-RII, and oxLDL increased significantly in the COVID survivors group compared to the group who never had COVID". However from the findings depicted in table 2 and the subsequent analysis the reader understands the exact oposite, which is also stated in the 'results' section; "Comparing the changes in markers between groups, COVID infection had a negative effect on inflammation VCAM, TNF-RI, TNF-RII..". The authors should clarify their findings and discuss them in relation to the findings of the study by Schultheiss et al. (Schultheiß C, Willscher E, Paschold L, Gottschick C, Klee B, Henkes SS, Bosurgi L, Dutzmann J, Sedding D, Frese T, Girndt M, Höll JI, Gekle M, Mikolajczyk R, Binder M. The IL-1β, IL-6, and TNF cytokine triad is associated with post-acute sequelae of COVID-19. Cell Rep Med. 2022 Jun 21;3(6):100663. doi: 10.1016/j.xcrm.2022.100663.).
Answer 1: By stating that it negatively affects inflammation, we intended to convey that this leads to worsened inflammation, reflected by higher inflammatory markers. However, we have now revised this section in the results for better clarity and substituted: ‘had a negative effect on ‘by ‘Increased the’.
To compare our results with the result of the mentioned study we added the following: Addionally, the result of a study done in Germany, reported chronic elevation in certain inflammatory marker among people with post-acute sequalae of SARS-CoV-2, which is consistent with the results of our study.
we added the reference.
2) The authors should provide thourough details about the mixed effect models (at least in a supplementary file).
answer 2: Thank you for comments. We added a few additional words/phrases in the statistical analysis section to provide additional clarity and do not feel there is any additional information to provide than what is already explained in the methods section.
3) The authors should clarify whether paired statistical tests were performed in order to identify whether changes in time (pre- and post-pamdemic differences) regarding the separate groups (COVID and non-COVID) were statistically significant. If this is not the case, the analysis should be repeated using paired tests.
Answer 3:The longitudinal models calculated the within subject changes. Additional language was added in the statistical analysis section for clarity.
4) In the "limitations" paragraph the authors should state that the EndoPAT method evaluates arterial stiffness indirectly.
answer 4: We added to the limitation section the following: the fact that the EnpoPAT method indirectly measures the arterial stiffness
5) The authors should explain in the 'limitations' paragraph how does the increased percentage of current smokers in the non-COVID group (compared to the COVID group) affect their analysis.
answer 5: Thank you for the comment. We do not, however, see this as a limitation. Although the COVID negative group had a larger proportion of current smokers, there was insufficient evidence to suggest smoking had any association or increased risk of having worse AI in our study. To shed light on this, we added a sentence in the results section.
6) Since high plasma osteopontin has been associated with atherosclerosis (Kadoglou NPE, Khattab E, Velidakis N, Gkougkoudi E. The Role of Osteopontin in Atherosclerosis and Its Clinical Manifestations (Atherosclerotic Cardiovascular Diseases)-A Narrative Review. Biomedicines. 2023 Nov 29;11(12):3178. doi: 10.3390/biomedicines11123178.) and hypercoagulability tends to increase atherosclerosis (Loeffen R, Spronk HM, ten Cate H. The impact of blood coagulability on atherosclerosis and cardiovascular disease. J Thromb Haemost. 2012 Jul;10(7):1207-16. doi: 10.1111/j.1538-7836.2012.04782.x.), the authors should discuss their findings in relation to previous reports indicating that patients with serious post-acute COVID-19 sequelae were characterized by increased plasma osteopontin levels (Pappas AG, Eleftheriou K, Vlahakos V, Magkouta SF, Riba T, Dede K, Siampani R, Kompogiorgas S, Polydora E, Papalampidou A, Loutsidi NE, Mantas N, Tavernaraki E, Exarchos D, Kalomenidis I. High Plasma Osteopontin Levels Are Associated with Serious Post-Acute-COVID-19-Related Dyspnea. J Clin Med. 2024 Jan 10;13(2):392. doi: 10.3390/jcm13020392.) and that the patients with long COVID were characterized by hypercoagulable ROTEM (Loutsidi NE, Politou M, Vlahakos V, Korakakis D, Kassi T, Nika A, Pouliakis A, Eleftheriou K, Balis E, Pappas AG, Kalomenidis I. Hypercoagulable Rotational Thromboelastometry During Hospital Stay Is Associated with Post-Discharge DLco Impairment in Patients with COVID-19-Related Pneumonia. Viruses. 2024 Dec 14;16(12):1916. doi: 10.3390/v16121916.). Â
answer 6: Thank you for this valuable suggestion, we added the following ‘in the context of inflammation, osteopontin stands out as a potential key biomarker that warrants further investigation, especially given its association with patients experiencing post-acute sequelae of COVID-19. Since long COVID is linked to a heightened coagulable state in the acute hospital stay. It seems that this osteopontin and hypercoagulability plays a significant role in the long-term sequelae observed after COVID-19, one of which is the worsening of arterial function post-infection seen in our study.’
We added the following two references suggested by the reviewer:
- Pappas AG, Eleftheriou K, Vlahakos V, Magkouta SF, Riba T, Dede K, Siampani R, Kompogiorgas S, Polydora E, Papalampidou A, et al. High Plasma Osteopontin Levels Are Associated with Serious Post-Acute-COVID-19-Related Dyspnea. Journal of Clinical Medicine. 2024; 13(2):392. https://doi.org/10.3390/jcm13020392
- Loutsidi NE, Politou M, Vlahakos V, et al. Hypercoagulable Rotational Thromboelastometry During Hospital Stay Is Associated with Post-Discharge DLco Impairment in Patients with COVID-19-Related Pneumonia. Viruses. 2024;16(12):1916. Published 2024 Dec 14. doi:10.3390/v16121916
Apart from a few grammar/vocabulary errors the manuscript is generally well written
Reviewer 3 Report
Comments and Suggestions for Authors
Baissary et al assessed the endothelial health in COVID-19 survivors and Never-COVID-19 patients in prepandemic and postpandemic era. They used Endopat for measuring AI and also measured several inflammatory markers. They concluded that AI index is worsened in COVID-19 survivors even some of them are mildly infected.
The strength of this study is homogenous cohort those are compared. Also they used 15 months after COVID-19 recovery for measuring AI. The weakness is patient numbers are not very high and it is a single center study.
The conclusion of this study is of importance. However, some concerns that need to be addressed.
- Authors followed up COVID-19 survivors until 15 months. Whether these patients had other non-cardiovascular symptoms (CDC or WHO defined) of Long COVID? Authors may put a paragraph in discussion regarding this that arterial stiffness is or not an isolated phenomenon in COVID survivors.
- The study period is July, 2021 to May, 2024. How many proportion of COVID survivors were single/double/booster or not vaccinated? Did authors perform any analysis considering vaccination and non-vaccination? As the vaccinated COVID-19 patients mostly have mild infections, whether AI analysis could be subdivided into mild and severe.
- In Table 1,. In pre-pandemic stage, it seems the AI value in NON-COVID group is very high (10.1+/- 16.6). This is supposed to be normal population and why is it so high AI? Authors should put a note in the text with explanation.
- In table 2, Authors should specify that the values in RHI<1.67 are patient no.
- In table 2, it is not understood why hsCRP is increased in NON-COVID patient group.
- In table 2, is the increase of sCD14 (186.1 vs -24.4) a feature of Long COVID? If so, authors should discuss it in the discussion with reference.
Author Response
Thank you for your insightful comments on my manuscript; they are greatly appreciated and have helped improve the work.
Baissary et al assessed the endothelial health in COVID-19 survivors and Never-COVID-19 patients in prepandemic and postpandemic era. They used Endopat for measuring AI and also measured several inflammatory markers. They concluded that AI index is worsened in COVID-19 survivors even some of them are mildly infected.
The strength of this study is homogenous cohort those are compared. Also they used 15 months after COVID-19 recovery for measuring AI. The weakness is patient numbers are not very high and it is a single center study.
The conclusion of this study is of importance. However, some concerns that need to be addressed.
1.Authors followed up COVID-19 survivors until 15 months. Whether these patients had other non-cardiovascular symptoms (CDC or WHO defined) of Long COVID? Authors may put a paragraph in discussion regarding this that arterial stiffness is or not an isolated phenomenon in COVID survivors.
Â
Response 1: Thank you for your comment. We found no significant association between arterial stiffness and other Long COVID symptoms, but we focused our report on only the impact of COVID on arterial stiffness and inflammation; to enhance the analysis and be able to make meaningful conclusion as suggested by the reviewer, a larger population would be required.
Â
2.The study period is July, 2021 to May, 2024. How many proportion of COVID survivors were single/double/booster or not vaccinated? Did authors perform any analysis considering vaccination and non-vaccination? As the vaccinated COVID-19 patients mostly have mild infections, whether AI analysis could be subdivided into mild and severe.
Â
Response  2: We do not have reliable information on vaccinations as most participants in our health system receive these vaccinations at outside drug stores or health systems unrelated to our EHR. The study did not capture vaccination status, although agree with reviewer that in retrospect, we should have. This is very important variable that we are including in our future research.
Â
3.In Table 1. In pre-pandemic stage, it seems the AI value in NON-COVID group is very high (10.1+/- 16.6). This is supposed to be normal population and why is it so high AI? Authors should put a note in the text with explanation.
Â
Response 3:Yes, we did notice that the values for AI are higher in the COVID-negative group. Groups had the endoPAT performed in a similar standardized manner and using the same machine. Please remember that at the pre-pandemic timepoint (Table-1), the entire study population is COVID-negative and that at this time point, there was insufficient evidence (i.e., P>0.05) to suggest that AI was different between the groups. Additionally in the analysis we looked into the proportion of change in AI.
Â
4.In table 2, Authors should specify that the values in RHI<1.67 are patient no.
Â
Response 4:Thank you. In table 2, at the top and underneath the groups we have mean ± std or % that describes the values in the table. We’ve added a footnote in the table to make it clearer that % represents a proportion.
Â
5.In table 2, it is not understood why hsCRP is increased in NON-COVID patient group.
Â
Response 5: Yes, the behavior of hsCRP is not what we might expect. One possible explanation is that the COVID-negative group had nearly two-times the proportion of current smokers. This may be an explanation for the increase in CRP but we could not be certain.
Â
Â
6.In table 2, is the increase of sCD14 (186.1 vs -24.4) a feature of Long COVID? If so, authors should discuss it in the discussion with reference.
Â
Response 6: Thank you for your comments. Although there was evidence (P<0.05) of an increase in sCD14 within the COVID positive group and a decrease in the COVID negative group between pre- and post-pandemic visits, our data suggests that this change (as with VCAM, TNFr-I/, and TNFr-II) is attributed to COVID-infection. Monocyte activation (sCD14) is a likely source of the heightened inflammation in chronic illnesses, like HIV and COVID.
Reviewer 4 Report
Comments and Suggestions for Authors
"Long COVID" or "post-COVID-19 condition", as the World Health Organization officially calls it, is defined as "the persistence or development of new symptoms for 3 months after initial SARS-CoV-2 infection, with symptoms persisting for at least 2 months without other explanation". Currently, there is no doubt that Long COVID is associated with the onset or progression of cardiovascular disease. In this regard, the data presented by the authors of the paper are relevant. Meanwhile, I have a number of comments regarding the quality of this paper:
(1) The groups without COVID and those with COVID were not completely comparable at the initial stage (before the COVID-19 epidemic) of the prospective study, such parameters as tobacco smoking (P=0.01), inflammatory markers: TNF-RI (P=0.03), TNF-RII (P<0.001) and I-CAM (P<0.001) - line 193. This circumstance imposes additional limitations on the evaluation of the results, especially with regard to the parameters of inflammation, which the authors should point out in the relevant section of the article.
(2) 2.3. Statistical analysis: "Characteristics of the study participants were described using mean +/- standard deviation (SD) or median and interquartile range (IQR) for continuous variables and frequency (n) and percentage (%) for categorical variables" The mean (M) and median (Me) estimates are used for normal and abnormal distributions, respectively, but it is not clear what method the authors used to determine the normality of the distribution in the groups.
(3) The soluble forms of the receptors should be reported as sTNF-RI, sTNF-RII, etc.
(4) VCAM and ICAM are families of adhesion receptors (mainly localized on endotheliocytes). The authors probably examined the concentrations of sVCAM-1 (sCD106) and sICAM-1 (sCD54) in blood plasma, which are commonly used in such studies.
(5) Tables 2 and 3 are presented as a photo, not as PDF text. In Table 1, it is desirable to note the values of p<0.05.
(6) It is desirable to present the comparative results of a prospective study conducted 15 months after the onset of acute COVID-19 and the control group at the appropriate time, specifically in a manner similar to the data in Table 1.
(7) sCD14 (presepsin) and sCD163 (scavenger receptor) are more likely markers of activation of stromal (mainly vascular) macrophages rather than monocytes, or if you want to write in tables: monocytes/macrophages.
(8) BDG (1-3-beta-D-glucan) is a polysaccharide of the wall of many pathogenic fungi, therefore the use of this indicator as a marker of intestinal permeability is only possible if a fungal infection is excluded.
(9) References should be adapted to MDPI style.
Author Response
Thank you for your insightful comments on my manuscript; they are greatly appreciated and have helped improve the work.
"Long COVID" or "post-COVID-19 condition", as the World Health Organization officially calls it, is defined as "the persistence or development of new symptoms for 3 months after initial SARS-CoV-2 infection, with symptoms persisting for at least 2 months without other explanation". Currently, there is no doubt that Long COVID is associated with the onset or progression of cardiovascular disease. In this regard, the data presented by the authors of the paper are relevant. Meanwhile, I have a number of comments regarding the quality of this paper:
(1) The groups without COVID and those with COVID were not completely comparable at the initial stage (before the COVID-19 epidemic) of the prospective study, such parameters as tobacco smoking (P=0.01), inflammatory markers: TNF-RI (P=0.03), TNF-RII (P<0.001) and I-CAM (P<0.001) - line 193. This circumstance imposes additional limitations on the evaluation of the results, especially with regard to the parameters of inflammation, which the authors should point out in the relevant section of the article.
Response 1: Thank you for the comment. As you noted, there were differences that may impose additional limitations on the evaluation of the results. This is expected in an observational non-randomized study. These differences were taken into consideration and controlled for in adjusted models. We’ve confirmed that this has been addressed in the methods section of the text.
(2) 2.3. Statistical analysis: "Characteristics of the study participants were described using mean +/- standard deviation (SD) or median and interquartile range (IQR) for continuous variables and frequency (n) and percentage (%) for categorical variables" The mean (M) and median (Me) estimates are used for normal and abnormal distributions, respectively, but it is not clear what method the authors used to determine the normality of the distribution in the groups.
Response 2: We’ve added additional information:  The Shapiro-Wilk test was used to test the normality assumption. Differences between groups were computed using an independent t-test or non-parametric, Mann-Whitney U Test, for continuous variables and chi-square or Fisher’s exact for categorical variables.
‎(3) The soluble forms of the receptors should be reported as sTNF-RI, sTNF-RII, etc.
Response 3: Thank you, we added it.
(4) VCAM and ICAM are families of adhesion receptors (mainly localized on endotheliocytes). The authors probably examined the concentrations of sVCAM-1 (sCD106) and sICAM-1 (sCD54) in blood plasma, which are commonly used in such studies.
Response 4: Thank you, we added it.
(5) Tables 2 and 3 are presented as a photo, not as PDF text. In Table 1, it is desirable to note the values of p<0.05.
Response 5: This question/statement is not clear to us what the reviewer is asking. We have noted the P-values in the footnote of Table-1. If our response has not addressed this comment, please clarify the question and we will respond appropriately.
(6) It is desirable to present the comparative results of a prospective study conducted 15 months after the onset of acute COVID-19 and the control group at the appropriate time, specifically in a manner similar to the data in Table 1.
Response 6: This question/statement is not clear to us what the reviewer is asking. For context and attempt to answer the intended question, the data presented in Table-1 is parsed out into two groups to describe similarities/differences in these two groups pre-pandemic (when both groups are technically COVID-negative), standard for cohort studies. Presenting the data in this manner also serves as a reference for Table-2, which is also separated by COVID-negative/positive groups, when observing the within-group changes at the follow-up post-pandemic visit. Table-2, intentionally, presents the within-group and between-group change separately to show the magnitude and direction of the change (which can be referenced then with Table-1).
If our response has not addressed this comment, please clarify the question and we will respond appropriately.
(7) sCD14 (presepsin) and sCD163 (scavenger receptor) are more likely markers of activation of stromal (mainly vascular) macrophages rather than monocytes, or if you want to write in tables: monocytes/macrophages.
Response 7: Thank you, added.
(8) BDG (1-3-beta-D-glucan) is a polysaccharide of the wall of many pathogenic fungi, therefore the use of this indicator as a marker of intestinal permeability is only possible if a fungal infection is excluded.
Response 8:Yes, all active infections were considered exclusionary for the study.
(9) References should be adapted to MDPI style.
Response 9: Updated.
Round 2
Reviewer 2 Report
Comments and Suggestions for Authors
The authors proceeded to the necessary improvements. The manuscript is -in my opinion- suitable for scientific publication.
Author Response
Thank you for your prompt response.
Reviewer 3 Report
Comments and Suggestions for Authors
Authors provided satisfactory answers for my concerns. But they did not modify the text accordingly. I want them to write these responses each with at least one sentence in discussion or in limitations (response 1, not associated with other long Covid symptoms), 2 (vaccine information is not available), 3(hsCRP value in non COVID) and 5, (CD14).
Author Response
Thanks again for your review and your comment.
We added the following to the discussion:
- There was evidence (P<0.05) of an increase in sCD14 within the COVID positive group and a decrease in the COVID negative group between pre- and post-pandemic visits, our data suggests that this change (as with VCAM-1, TNFr-I, and TNFr-II) is attributed to COVID-infection. Monocyte activation (sCD14) is a likely source of the heightened inflammation in chronic illnesses. Our results also showed a non-significant increase in hsCRP in the COVID negative group, but one possible explanation is that the COVID-negative group had nearly two-times the proportion of current smokers.
- Our study showed no significant correlation between the arterial function changes and other long COVID symptoms.
We added the following to the limitation section:
- Although we lacked sufficient data on the vaccination status of the participants, this variable is crucial and should be considered in future research.
Reviewer 4 Report
Comments and Suggestions for Authors
(4) VCAM and ICAM are families of adhesion receptors (mainly localized on endotheliocytes). The authors probably examined the concentrations of sVCAM-1 (sCD106) and sICAM-1 (sCD54) in blood plasma, which are commonly used in such studies.
Response 4: Thank you, we added it.
However, upon careful examination, these alterations remain elusive in tables 1 and 2. It is imperative to reiterate the following assertion: the ICAM family of adhesion receptors encompasses ICAM-1, ICAM-2, and ICAM-3, while the VCAM family in humans comprises VCAM-1 and VCAM-2. Consequently, in Tables 1 and 2, the authors are obliged to explicitly specify which molecules from the VCAM and ICAM families they have investigated.
Author Response
Thank you for your comment.
Yes, that's right, in our study we examined the ICAM1 and VCAM1 and I added that to the manuscript.Â
Â
Thanks again